# MoRA: Missing Modality Low-Rank Adaptation for Visual Recognition

**Shu Zhao[1], Nilesh Ahuja[2], Tan Yu[3], Tianyi Shen[1], Vijaykrishnan Narayanan[1]**
[1]The Pennsylvania State University [2]Intel [3]NVIDIA
smz5505@psu.edu

## Abstract

Pre-trained vision language models have shown remarkable performance on visual recognition tasks, but they typically assume the availability of complete multimodal inputs during both training and inference. In real-world scenarios, however, modalities may be missing due to privacy constraints, collection difficulties, or resource limitations. While previous approaches have addressed this challenge using prompt learning techniques, they fail to capture the cross-modal relationships necessary for effective multimodal visual recognition and suffer from inevitable computational overhead. In this paper, we introduce MoRA, a parameter-efficient fine-tuning method that explicitly models cross-modal interactions while maintaining modality-specific adaptations. MoRA introduces modality-common parameters between text and vision encoders, enabling bidirectional knowledge transfer. Additionally, combined with the modality-specific parameters, MoRA allows the backbone model to maintain inter-modality interaction and enable intra-modality flexibility. Extensive experiments on standard benchmarks demonstrate that MoRA achieves an average performance improvement in missing-modality scenarios by $5.24\%$ and uses only $25.90\%$ of the inference time compared to the SOTA method while requiring only $0.11\%$ of trainable parameters compared to full fine-tuning. The code is available at `https://github.com/Tree-Shu-Zhao/MoRA`.

## 1 Introduction

Pre-trained vision language models (VLMs) integrate multiple modalities (e.g., vision and language) to comprehensively understand their environment, demonstrating remarkable performance on various downstream tasks, including visual recognition (Hu et al., 2024) and cross-modal retrieval (Li et al., 2025). VLMs like CLIP (Radford et al., 2021) and ViLT (Kim et al., 2021) leverage large-scale paired data to learn joint representations of images and text. Multimodal large language models, including GPT-4 (Achiam et al., 2023), Gemini (Team et al., 2024), LLaMA-Vision (Grattafiori et al., 2024), and LLaVA (Liu et al., 2023a), build connections between vision and language and use the knowledge within LLMs to establish powerful conversation and reasoning abilities.

Despite their impressive capabilities, deploying them in real-world scenarios presents two significant challenges. First, most multimodal models work under the assumption of modality completeness, requiring all modalities to be available during both training and inference. However, this assumption rarely holds in practice due to privacy constraints, collection difficulties, or resource limitations (Ma et al., 2022). When input modalities are missing, performance degrades substantially (Hu et al., 2024), limiting their applicability in real-world settings where data completeness cannot be guaranteed. Second, as model sizes grow, fine-tuning becomes increasingly computationally expensive with limited resources and leads to overfitting on small-scale target datasets (Khattak et al., 2023). Although several works (Lee et al., 2023; Hu et al., 2024) have devised prompt-based methods to alleviate them, the prompts lead to inevitable inference overhead.

To address these challenges, we explore the underlying mechanisms affecting the performance when modalities are missing. A critical insight comes from Mind the Gap (Liang et al., 2022), identifying the "modality gap" which is the geometric separation between different modality embeddings in the shared representation space. Building on this observation, we argue that both the alignment and gap

between modalities provide valuable complementary information for improving performance during inference with missing modalities. Specifically, during fine-tuning, the embedding spaces of the visual and text encoders should be related, moving in the same direction to maintain multimodal performance. Simultaneously, these encoders need to maintain their own independent update directions to better adapt to downstream tasks without compromising modality-specific characteristics.

Inspired by this, we introduce MoRA, a parameter-efficient fine-tuning method that explicitly models cross-modal interactions while maintaining modality-specific adaptations. MoRA incorporates two key design elements: a shared cross-modal parameter module that enables knowledge transfer between modalities through the Gram matrix (Strang, 2022) of shared low-rank parameters and modality-specific adaptation components that preserve the unique characteristics of each modality. This dual-structure design allows MoRA to maintain inter-modality interactions while enabling intra-modality flexibility, resulting in robust performance across various missing-modality scenarios.

To summarize our contributions, we propose MoRA, a parameter-efficient fine-tuning method for multimodal models that explicitly addresses the challenge of missing modalities through shared cross-modal parameters and modality-specific adaptations, enabling bidirectional knowledge transfer between modalities while preserving the directional properties of the original weights. We design an efficient training strategy that requires updating only a small fraction ($\sim 0.11\%$) of the model parameters, making it feasible to adapt large pre-trained models even with limited computational resources. Through extensive experiments on standard benchmarks, we demonstrate that MoRA significantly outperforms existing prompt-based and parameter-efficient approaches across various missing-modality scenarios while maintaining inference efficiency.

## 2    RELATED WORK

### 2.1    MISSING MODALITY FOR MULTIMODAL LEARNING

The missing modality issue presents a significant challenge in deploying robust systems, leading to a significant performance drop. Previous approaches for addressing missing modality challenges can be broadly categorized into Alignment-based and Reconstruction-based methods. Alignment-based methods (Wang et al., 2023; Zhang et al., 2023b; Shvetsova et al., 2022) embed different modalities into a shared representation space, enabling the model to operate effectively even when certain modalities are missing by aligning the feature spaces of different modalities during pre-training or fine-tuning. Reconstruction-based methods (Ma et al., 2022; Zhao et al., 2021; Ma et al., 2021) use available modalities to reconstruct features of missing modalities explicitly. These approaches typically employ generative models or cross-modal translation networks to synthesize the absent information, allowing the model to operate on "completed" inputs. However, these methods often suffer from imperfect reconstruction quality, especially when the missing modality contains information that cannot be fully inferred from available ones. More recently, prompt learning techniques (Lee et al., 2023; Hu et al., 2024) have emerged as a subset of reconstruction-based approaches, handling missing-modality scenarios by inserting learnable tokens into transformer layers. Modality-specific information is offloaded to learnable prompts and reused when modalities are missing. MMP (Lee et al., 2023) treats different missing-modality cases as different types of input, adapting the model through learnable prompts while keeping the backbone frozen. However, MMP inserts independent prompts into each layer, overlooking the complex relationships between modalities. DCP (Hu et al., 2024) and SyP (Zhang et al., 2025) devise more prompts to leverage the correlations between prompts and input features across different layers. However, it discards the features of the missing modalities and cannot fully exploit multimodal features for downstream tasks. MoRA preserves the modality information during training and introduces no overhead during inference.

### 2.2    PARAMETER-EFFICIENT FINE-TUNING

Parameter-Efficient Fine-Tuning (PEFT) methods reduce the computational burden of adapting large models by updating only a small subset of parameters. These approaches can be classified into three categories. Adapter-based methods (Houlsby et al., 2019) insert trainable modules into backbones, either sequentially or in parallel with existing layers. Prompt-based methods (Liu et al., 2023b) add trainable tokens to the input while keeping model parameters fixed. Both categories typically introduce additional inference latency. Low-Rank Adaptation (LoRA) methods (Hu et al., 2022) approx-

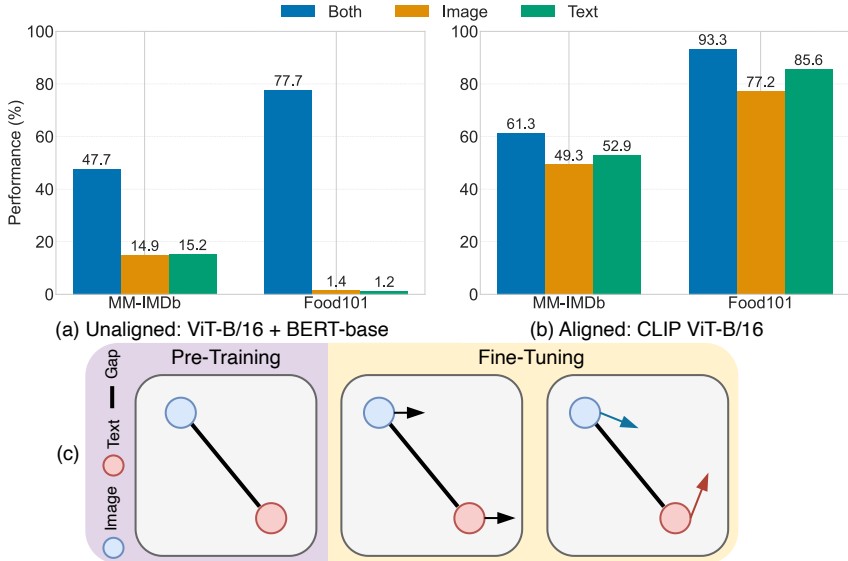

Figure 1: Motivation for MoRA. (a) Performance comparison on MM-IMDb and Food101 datasets using unaligned vision and text encoders. (b) Performance comparison using aligned CLIP ViT-B/16 encoder. (c) During pre-training, modalities are aligned in embedding space with a gap; during fine-tuning, modalities should maintain their relationship while allowing modality-specific adaptations.

imate weight updates using low-rank matrices that can be merged with pre-trained weights before inference, thus maintaining inference efficiency. Various extensions have been proposed, including SVD-based approaches (Zhang et al., 2023a), orthogonal factorization (Qiu et al., 2023; Liu et al., 2024b), and direction decomposition (Liu et al., 2024a). While Multimodal LoRA methods (Shen et al., 2024; Ge et al., 2025) have focused on instruction tuning, they cannot handle missing modalities and lack architectural innovations for cross-modal interaction in dual-branch architectures. Shi et al. (2024) address the task in medical diagnosis through unidirectional adaptation. MoRA targets general visual recognition tasks with bidirectional knowledge transfer, achieving superior efficiency with smaller trainable parameters and zero inference latency.

## 3 METHOD

### 3.1 PROBLEM FORMULATION

We focus on the multimodal classification task with missing modalities during both training and testing. For simplicity, but without loss of generality, we consider a multimodal dataset with text ($t$) and vision ($v$) modalities, i.e., $\mathcal{D} = \{\mathcal{D}^t, \mathcal{D}^v, \mathcal{D}^c\}$. Specifically, $\mathcal{D}^t = \{(\mathbf{t}_i, \mathbf{y}_i)\}_{i=1}^{N_t}$ contains text-only data samples; $\mathcal{D}^v = \{(\mathbf{v}_i, \mathbf{y}_i)\}_{i=1}^{N_v}$ includes image-only data samples; $\mathcal{D}^c = \{(\mathbf{t}_i, \mathbf{v}_i, \mathbf{y}_i)\}_{i=1}^{N_c}$ is the subset containing modality-complete samples with both text and image, where $\mathbf{t}_i$ is text, $\mathbf{v}_i$ denotes an image, and $\mathbf{y}_i \in \mathbb{R}^C$ is the label vector where $C$ is the number of classes. When the image is missing, we set the image input to an all-1 matrix; when the text is missing, we set the text input to an empty string.

### 3.2 MOTIVATION

Vision Language Models (VLMs) have been pre-trained on massive image-text pairs. Although the pre-training stage aligns the vision and language embedding space, Mind the Gap (Liang et al., 2022) points out that there is still a gap between modalities. We argue that both the alignment property and the gap are important for the missing modality task. To demonstrate these, we fine-tune aligned and unaligned models using modality-complete samples and test them using both complete and incomplete samples, as illustrated in Figure 1 (a) (b). The aligned/unaligned models denote whether vision and text encoders are trained on image-text pairs. Implementation details can be found in

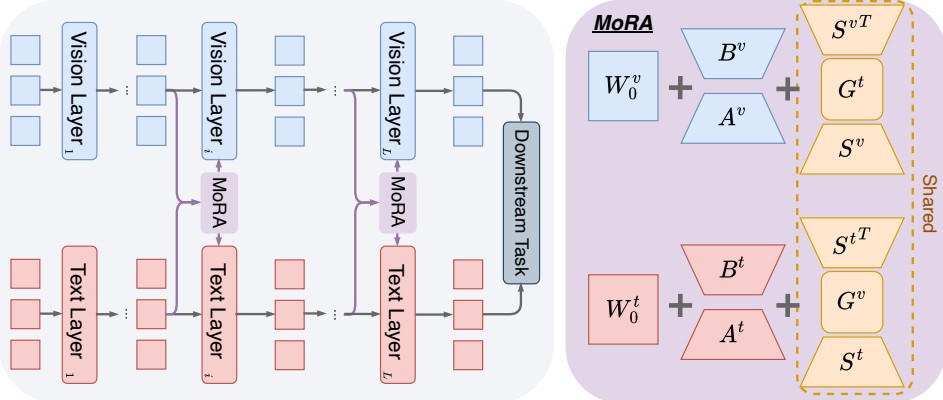

Figure 2: Overview of the proposed MoRA architecture.

Section A.3. Compared to the performance drop of $-11.1$ using the aligned model, the unaligned model shows a drop of $-54.5$, demonstrating that other available aligned modalities can maintain a certain level of performance when modalities are missing. Additionally, using both image and text features with the aligned model achieves better performance than using only image features ($-14.1$) or only text features ($-11.9$). This finding suggests that the gap represents different information across modalities, which serves as important complementary information for multimodal tasks.

Therefore, we identify two properties that need to be considered during fine-tuning VLMs, as illustrated in Figure 1 (c). First, the direction of fine-tuning image and text modalities should be the same to maintain their relationship in embedding space for general ability. Second, the image and text modalities should have their own fine-tuning direction to enable flexibility for downstream tasks. Inspired by these, we propose MoRA, a parameter-efficient fine-tuning method that explicitly models cross-modal interactions while maintaining modality-specific adaptations.

### 3.3 MISSING MODALITY LOW-RANK ADAPTATION

The weight matrix in LoRA can be decomposed into the magnitude and direction, as shown below:

$$\mathbf{W} = \mathbf{W}_0 + \Delta\mathbf{W} = \|\mathbf{W}_0 + \Delta\mathbf{W}\|_{\mathrm{F}} \frac{\mathbf{W}_0 + \Delta\mathbf{W}}{\|\mathbf{W}_0 + \Delta\mathbf{W}\|_{\mathrm{F}}} = \|\mathbf{W}_0 + \Delta\mathbf{W}\|_{\mathrm{F}} \overline{\mathbf{W}_0 + \Delta\mathbf{W}}, \quad (1)$$

where $\|\mathbf{W}\|_{\mathrm{F}}$ is the Frobenius norm of the matrix, denoting the magnitude; $\overline{\mathbf{W}}$ is the normalized matrix, denoting the direction.

Although recent works (Liu et al., 2024a; Wu et al., 2025) have shown the importance of the direction in fine-tuning models, they focus on large language models, while the cross-modality information interaction, which is important for multimodal tasks, is not discussed. Based on the analysis in Section 3.2, we introduce MoRA, a parameter-efficient fine-tuning method that enables cross-modal interactions and captures modality-common/specific information during training, as illustrated in Figure 2. MoRA introduces two types of learnable parameters, including modality-specific parameters $\mathbf{A}^{\mathrm{v/t}} \in \mathbb{R}^{r \times d_{\mathrm{v/t}}}, \mathbf{B}^{\mathrm{v/t}} \in \mathbb{R}^{d_{\mathrm{v/t}} \times r}$ for independent adaptation, and shared parameters $\mathbf{S}^v \in \mathbb{R}^{r \times d_{\mathrm{v}}}, \mathbf{S}^t \in \mathbb{R}^{d_{\mathrm{t}} \times r}$ for cross-modal knowledge transfer, where $d_{\mathrm{v}}$ and $d_{\mathrm{t}}$ are the dimensions of vision and text encoders respectively, and $r \ll d$ is the rank. The updated weight matrix for image (v) / text (t) encoders is:

$$
\begin{aligned}
\mathbf{W}^{\mathrm{v/t}} &= \mathbf{W}_0^{\mathrm{v/t}} + \Delta\mathbf{W}^{\mathrm{v/t}} + \Delta\mathbf{W}^{\mathrm{s}} \\
&= \left(\mathbf{W}_0^{\mathrm{v/t}} + \Delta\mathbf{W}^{\mathrm{v/t}}\right) + \left(\mathbf{W}_0^{\mathrm{v/t}} + \Delta\mathbf{W}^{\mathrm{s}}\right) \cancel{-\mathbf{W}_0^{\mathrm{v/t}}} \quad -\mathbf{W}_0^{\mathrm{v/t}} \text{ is frozen and ignored} \\
&= \|\mathbf{W}_0^{\mathrm{v/t}} + \Delta\mathbf{W}^{\mathrm{v/t}}\|_{\mathrm{F}} \frac{\mathbf{W}_0^{\mathrm{v/t}} + \Delta\mathbf{W}^{\mathrm{v/t}}}{\|\mathbf{W}_0^{\mathrm{v/t}} + \Delta\mathbf{W}^{\mathrm{v/t}}\|_{\mathrm{F}}} + \|\mathbf{W}_0^{\mathrm{v/t}} + \Delta\mathbf{W}^{\mathrm{s}}\|_{\mathrm{F}} \frac{\mathbf{W}_0^{\mathrm{v/t}} + \Delta\mathbf{W}^{\mathrm{s}}}{\|\mathbf{W}_0^{\mathrm{v/t}} + \Delta\mathbf{W}^{\mathrm{s}}\|_{\mathrm{F}}} \quad (2) \\
&= \|\mathbf{W}_0^{\mathrm{v/t}} + \Delta\mathbf{W}^{\mathrm{v/t}}\|_{\mathrm{F}} \overline{\mathbf{W}_0^{\mathrm{v/t}} + \Delta\mathbf{W}^{\mathrm{v/t}}} + \|\mathbf{W}_0^{\mathrm{v/t}} + \Delta\mathbf{W}^{\mathrm{s}}\|_{\mathrm{F}} \overline{\mathbf{W}_0^{\mathrm{v/t}} + \Delta\mathbf{W}^{\mathrm{s}}} \\
&= \underbrace{\alpha^{\mathrm{v/t}} \overline{\mathbf{W}_0^{\mathrm{v/t}}} + \mathbf{B}^{\mathrm{v/t}} \mathbf{A}^{\mathrm{v/t}}}_{\text{Modality-Specific}} + \underbrace{\alpha^{\mathrm{s}} \overline{\mathbf{W}_0^{\mathrm{v/t}}} + \mathbf{S}^{\mathrm{v/t}} \mathbf{S}^{\mathrm{t/v}}}_{\text{Modality-Shared}},
\end{aligned}
$$

where $\alpha^{\mathrm{v/t}}$ and $\alpha^{\mathrm{s}}$ denote the learnable modality-specific and modality-shared magnitudes; $\mathbf{W}_0^{\mathrm{v/t}}$ is the frozen pre-trained weights in vision/text encoders.

However, Equation (2) only works when the dimensions of the image and text encoders are the same. For example, the dimension of vision ($d_{\mathrm{v}}$) and text encoders ($d_{\mathrm{t}}$) in the CLIP ViT-B/16 model is 768 and 512, i.e., $\mathbf{W}_0^{\mathrm{v}} \in \mathbb{R}^{768 \times 768}$ and $\mathbf{W}_0^{\mathrm{t}} \in \mathbb{R}^{512 \times 512}$. Direct multiplication of $\mathbf{S}^{\mathrm{v}}$ and $\mathbf{S}^{\mathrm{t}}$ would yield $\mathbb{R}^{r \times d_{\mathrm{v}}} \times \mathbb{R}^{r \times d_{\mathrm{t}}}$, which is incompatible with both encoder dimensions. This dimension mismatch challenge significantly limits the applicability of MoRA. Although we can add projection layers to map the image and text embeddings to a common space, the projection layers will significantly increase the number of learnable parameters during training and cannot be absorbed into the pre-trained weights $\mathbf{W}_0$, increasing the inference latency. We resolve this dimension mismatch by operating in the rank space through Gram matrices (Strang, 2022). For shared parameters $\mathbf{S}^{\mathrm{v}}$ and $\mathbf{S}^{\mathrm{t}}$, we compute:

$$\mathbf{G}^{\mathrm{v}} = \mathbf{S}^{\mathrm{v}}\mathbf{S}^{\mathrm{v}T} \in \mathbb{R}^{r \times r}$$
$$\mathbf{G}^{\mathrm{t}} = \mathbf{S}^{\mathrm{t}}\mathbf{S}^{\mathrm{t}T} \in \mathbb{R}^{r \times r}. \tag{3}$$

The key insight is that these Gram matrices capture the structural information of each modality in a dimension-agnostic rank space. We then use cross-modal Gram matrices to update each encoder:

$$\mathbf{W}^{\mathrm{v}} = \alpha^{\mathrm{v}}\overline{\mathbf{W}_0^{\mathrm{v}} + \mathbf{B}^{\mathrm{v}}\mathbf{A}^{\mathrm{v}}} + \alpha^{\mathrm{s}}\overline{\mathbf{W}_0^{\mathrm{v}} + \mathbf{S}^{\mathrm{v}T}\mathbf{G}^{\mathrm{t}}\mathbf{S}^{\mathrm{v}}}$$
$$\mathbf{W}^{\mathrm{t}} = \alpha^{\mathrm{t}}\overline{\mathbf{W}_0^{\mathrm{t}} + \mathbf{B}^{\mathrm{t}}\mathbf{A}^{\mathrm{t}}} + \alpha^{\mathrm{s}}\overline{\mathbf{W}_0^{\mathrm{t}} + \mathbf{S}^{\mathrm{t}T}\mathbf{G}^{\mathrm{v}}\mathbf{S}^{\mathrm{t}}}. \tag{4}$$

Since $\mathbf{S}^{\mathrm{v}T}\mathbf{G}^{\mathrm{t}}\mathbf{S}^{\mathrm{v}} \in \mathbb{R}^{d_{\mathrm{v}} \times d_{\mathrm{v}}}$ and $\mathbf{S}^{\mathrm{t}T}\mathbf{G}^{\mathrm{v}}\mathbf{S}^{\mathrm{t}} \in \mathbb{R}^{d_{\mathrm{t}} \times d_{\mathrm{t}}}$, they can be absorbed into the pre-trained weights during inference.

**Discussion**   First, the rank space captures second-order statistics of the low-rank representations, which extracting invariant representations across domains (Arjovsky et al., 2019). Second, the low-rank structure serves as a cross-modal adaptation module that transforms modality-specific parameters to incorporate shared knowledge (Srebro & Shraibman, 2005). With Gram matrices, MoRA maintains a balance between preserving modality-specific characteristics and enabling cross-modal information exchange. More importantly, all introduced learnable parameters can be absorbed into the original pre-trained weights, which makes **MoRA introduce no overheads during inference**.

# 4 EXPERIMENTS

## 4.1 EXPERIMENTAL SETUP

We evaluate MoRA on three benchmarks, including MM-IMDb (Ovalle et al., 2017), UPMC-Food101 (Wang et al., 2015), and Hateful Memes (Kiela et al., 2020). We adopt F1-Macro, top-1 classification accuracy, and Area Under the Receiver Operating Characteristic Curve (AUROC) to evaluate the three benchmarks, respectively. More details can be found in Section A.

**Missing Modality Setting**   We adopt a rigorous approach wherein modality absence occurs throughout both the training and inference phases. Following previous works (Lee et al., 2023; Hu et al., 2024), we designate $\eta\%$ as the missing ratio that quantifies the proportion of incomplete-modality data. In single-modality missing scenarios, the distribution follows a ratio of $\eta\%$ incomplete-modality samples to $1 - \eta\%$ complete-modality samples. When addressing dual-modality absences, the dataset consists of $\frac{\eta}{2}\%$ image-only instances and $\frac{\eta}{2}\%$ text-only instances, complemented by $1 - \eta\%$ of samples containing both modalities. This configuration effectively simulates real-world modality scarcity conditions and provides a robust framework for evaluating performance in missing modality environments. Implementation details can be found in Section A.

## 4.2 MAIN RESULTS

As shown in Table 1, MoRA consistently outperforms baseline methods across all missing ratio settings. Several key findings emerge from our experiments. First, the text modality consistently demonstrates greater importance than the image modality across all three datasets. This asymmetry may partially stem from text containing direct label information in certain datasets like UPMC-Food101. Second, MoRA achieves particularly strong improvements when the image modality is

Table 1: Performance comparison on MM-IMDb, Food101, and Hateful Memes datasets with varying missing ratios. MoRA consistently outperforms all baselines with average improvements of $5.30\%$, $1.91\%$, and $8.51\%$ over the next best method DCP.

| Datasets | $\eta$ | Image | Text | CoOp | MMP | MaPLe | DePT | DCP | MoRA |
|---|---|---|---|---|---|---|---|---|---|
| MM-IMDb | 50% | 100% | 50% | 48.06 | 48.88 | 49.58 | 50.64 | 52.13 | **54.62 (+2.49)** |
| | | 50% | 100% | 49.89 | 51.46 | 52.32 | 52.78 | 54.32 | **57.61 (+3.29)** |
| | | 75% | 75% | 48.37 | 49.32 | 49.56 | 50.87 | 52.32 | **55.88 (+3.56)** |
| | | *Average* | | 48.77 | 49.89 | 50.49 | 51.43 | 52.92 | **56.04 (+3.12)** |
| | 70% | 100% | 30% | 44.13 | 45.64 | 45.52 | 46.38 | 48.52 | **52.56 (+4.04)** |
| | | 30% | 100% | 48.82 | 50.52 | 50.64 | 52.13 | 53.14 | **56.39 (+3.25)** |
| | | 65% | 65% | 46.84 | 48.12 | 49.16 | 50.32 | 51.42 | **52.97 (+1.55)** |
| | | *Average* | | 46.60 | 48.09 | 48.44 | 49.61 | 51.03 | **53.97 (+2.94)** |
| | 90% | 100% | 10% | 44.76 | 45.32 | 46.84 | 47.56 | 49.26 | **50.67 (+1.41)** |
| | | 10% | 100% | 48.32 | 49.12 | 50.13 | 50.88 | 52.22 | **53.57 (+1.35)** |
| | | 55% | 55% | 44.12 | 44.87 | 45.12 | 46.54 | 48.04 | **51.64 (+3.60)** |
| | | *Average* | | 45.73 | 46.44 | 47.36 | 48.33 | 49.84 | **51.96 (+2.12)** |
| Food101 | 50% | 100% | 50% | 77.45 | 77.89 | 79.64 | 80.16 | 82.11 | **84.41 (+2.30)** |
| | | 50% | 100% | 87.02 | 87.16 | 87.35 | 82.14 | 89.12 | **89.63 (+0.51)** |
| | | 75% | 75% | 81.24 | 81.72 | 82.34 | 83.12 | 85.24 | **86.68 (+1.44)** |
| | | *Average* | | 81.90 | 82.26 | 83.11 | 81.81 | 85.49 | **86.91 (+1.42)** |
| | 70% | 100% | 30% | 76.34 | 76.52 | 77.02 | 77.34 | 78.87 | **80.85 (+1.98)** |
| | | 30% | 100% | 84.78 | 85.64 | 85.89 | 86.12 | 87.32 | **88.01 (+0.69)** |
| | | 65% | 65% | 78.87 | 79.12 | 79.84 | 81.46 | 81.87 | **83.77 (+1.90)** |
| | | *Average* | | 80.00 | 80.43 | 80.92 | 81.64 | 82.69 | **84.21 (+1.52)** |
| | 90% | 100% | 10% | 71.87 | 73.14 | 73.46 | 74.12 | 75.26 | **78.41 (+3.15)** |
| | | 10% | 100% | 81.67 | 82.14 | 83.12 | 83.56 | 85.78 | **86.77 (+0.99)** |
| | | 55% | 55% | 76.46 | 76.58 | 77.85 | 78.12 | 79.87 | **81.09 (+1.22)** |
| | | *Average* | | 76.67 | 77.29 | 78.14 | 78.60 | 80.30 | **82.09 (+1.79)** |
| Hateful Memes | 50% | 100% | 50% | 60.56 | 60.31 | 60.87 | 61.87 | 62.32 | **70.66 (+8.34)** |
| | | 50% | 100% | 62.41 | 62.35 | 63.13 | 63.88 | 64.46 | **71.58 (+7.12)** |
| | | 75% | 75% | 64.87 | 65.84 | 65.46 | 65.86 | 66.02 | **69.58 (+3.56)** |
| | | *Average* | | 62.61 | 62.83 | 63.15 | 63.87 | 64.27 | **70.61 (+6.34)** |
| | 70% | 100% | 30% | 60.74 | 61.12 | 61.26 | 61.56 | 62.82 | **69.43 (+6.61)** |
| | | 30% | 100% | 62.74 | 63.24 | 63.14 | 63.48 | 64.12 | **70.68 (+6.56)** |
| | | 65% | 65% | 64.82 | 65.04 | 65.23 | 65.48 | 66.08 | **70.15 (+4.07)** |
| | | *Average* | | 62.77 | 63.13 | 63.21 | 63.51 | 64.34 | **70.09 (+5.75)** |
| | 90% | 100% | 10% | 60.03 | 57.21 | 60.74 | 61.14 | 62.08 | **68.52 (+6.44)** |
| | | 10% | 100% | 61.46 | 61.52 | 61.87 | 62.42 | 63.87 | **68.78 (+4.91)** |
| | | 55% | 55% | 64.32 | 63.34 | 64.85 | 65.37 | 66.78 | **68.37 (+1.59)** |
| | | *Average* | | 61.94 | 60.69 | 62.49 | 62.98 | 64.24 | **68.56 (+4.32)** |

missing, highlighting its effectiveness in addressing the inadequate visual understanding of current methods through cross-modal knowledge interaction. Third, MoRA maintains remarkable robustness even under extreme conditions with a $90\%$ missing ratio on Hateful Memes, it achieves performance comparable to DCP at only $50\%$ missing ratio, demonstrating its superior ability to handle severe modality scarcity. These results validate that our dual mechanism of modality-specific adaptation and cross-modal parameter sharing creates a more resilient multimodal learning framework.

## 4.3 CROSS-SCENARIO GENERALIZATION

To evaluate the generalization capability of MoRA across different missing-modality scenarios, we conduct cross-scenario experiments where models are trained with one missing-modality configuration at a $70\%$ missing ratio and tested on different configurations. This evaluation is crucial for real-world deployment where the missing-modality patterns during inference may differ from those seen during training. We consider two training strategies: (1) training on both-missing scenarios where samples randomly have either text or image modality missing, and (2) training on single-modality scenarios where only one specific modality is consistently missing. We then evaluate these models on three test configurations: both-missing, image-missing, and text-missing scenarios.

As shown in Figure 3, MoRA demonstrates superior cross-scenario generalization compared to DCP across all configurations on the Hateful Memes dataset. When trained on both-missing scenarios, MoRA maintains strong performance when tested on specific missing-modality cases, signifi-

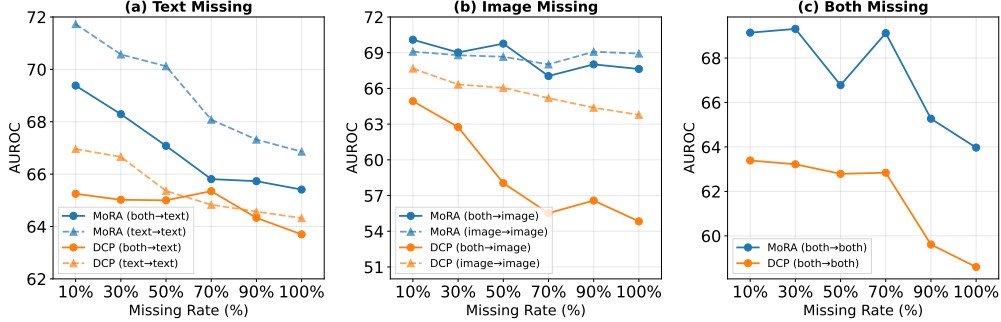

Figure 3: Generalizability Analysis on Hateful Memes dataset. (a) Models are trained on missing-both or missing-text cases, and evaluated on missing-text cases. (b) Models are trained on missing-both or missing-image cases, and evaluated on missing-image cases. (c) All models are trained on missing-both cases, and evaluated on missing-both cases.

Table 2: Inter-modal Distance Analysis. Average $L_2$ distance and angle between vision and text embeddings on Food101 test set.

| Method | $L_2$ Dist. | Angle (°) |
|---|---|---|
| CLIP (orig.) | 1.18 | 72.44 |
| FFT | 22.61 | 91.64 |
| DCP | 15.78 | 86.92 |
| **MoRA** | **9.99** | **77.07** |

Table 3: Modality-Specific Drift Analysis. Average embedding shift from original CLIP representations.

| Method | Vision | | Text | |
|---|---|---|---|---|
| | $L_2$ | Angle | $L_2$ | Angle |
| FFT | 8.36 | 92.16 | 20.60 | 87.67 |
| DCP | 8.22 | 65.17 | 13.57 | 66.26 |
| **MoRA** | **8.12** | **43.24** | **6.04** | **44.84** |

cantly outperforming DCP. This advantage persists even in challenging out-of-distribution scenarios—when models trained on text-missing data are tested on image-missing cases. The consistent performance gaps across all train-test combinations demonstrate that MoRA's dual mechanism of maintaining modality-specific parameters while enabling cross-modal knowledge transfer through Gram matrices creates a more robust representation space, particularly valuable for real-world deployments where missing-modality patterns may vary unpredictably from training conditions.

## 4.4 DIRECTION PROPERTY IN MoRA

To quantitatively validate our motivation illustrated in Figure 1(c), we conduct comprehensive analysis comparing the embedding space of different approaches. We train models with $70\%$ missing ratio where both modalities are absent, then evaluate on complete test samples to measure how well each method preserves inter-modal relationships while enabling adaptation.

**Inter-modal Relationship Preservation.** We measure the average $L_2$ distance and angle between vision and text embeddings for each category in the Food101 test set. As shown in Table 2, MoRA maintains the inter-modal distance and alignment with the original CLIP. In contrast, FFT, which fine-tunes all parameters, severely distorts these relationships, while DCP shows substantial degradation. This demonstrates that MoRA successfully preserves the aligned embedding structure crucial for handling missing modalities.

**Modality-Specific Adaptation.** We analyze the embedding drift from original CLIP representations to measure modality-specific flexibility. Table 3 shows that MoRA achieves balanced adaptation with minimal drift, significantly outperforming FFT which exhibits catastrophic drift. DCP shows moderate drift but fails to maintain the inter-modal alignment as shown above.

## 4.5 EIGENVALUE ANALYSIS OF MoRA

We conduct an eigenspectrum analysis of the Gram matrices used in MoRA and compare them to the pre-trained weights. We extract eigenvalues from the Gram matrices and singular values from

Table 4: Performance comparison of different parameter-efficient fine-tuning methods.

|  | MM-IMDb | Food101 | Hateful Memes |
|---|---|---|---|
| *Low-Rank-Based* | | | |
| MoRA | **52.97** | **83.77** | **70.15** |
| LoRA | 51.35 | 82.14 | 67.97 |
| DoRA | 51.89 | 82.34 | 68.28 |
| *Prompt-Based* | | | |
| DePT | 50.32 | 81.46 | 65.48 |
| DCP | 51.42 | 81.87 | 66.08 |
| *Weight Fine-Tuning* | | | |
| BitFit | 48.57 | 79.38 | 64.10 |
| FFT | 3.01 | 14.05 | 46.91 |

Table 5: Comparison with multimodal alignment and fusion methods.

|  | MM-IMDb | Food101 | Hateful Memes |
|---|---|---|---|
| MoRA | **52.97** | **83.77** | **70.15** |
| Align | 51.39 | 81.14 | 68.53 |
| Fusion | 50.72 | 81.01 | 68.17 |
| w/o Specific | 51.18 | 81.32 | 68.71 |
| w/o Gram | 50.41 | 80.31 | 68.19 |
| w/ Learnable Gram | 52.25 | 83.37 | 69.12 |
| w/ $\mathbf{W}_0$ | 52.88 | 83.59 | 70.03 |

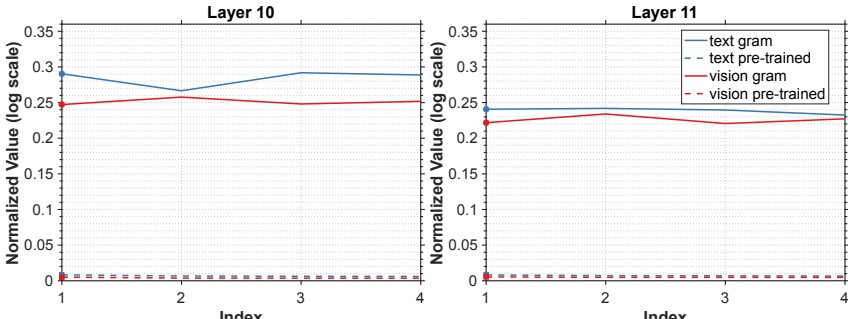

Figure 4: Comparison of eigenvalue distributions between Gram matrices and pre-trained weights.

the pre-trained weights in layers 10 and 11 of the vision and text encoders. Figure 4 presents the normalized eigenvalue distributions. Our analysis reveals several critical findings.

**First, Gram matrices serve as information concentration mechanisms**. This substantial difference demonstrates that Gram matrices effectively concentrate information in a much more compact form. To verify this empirically, we remove the Gram matrix, denoted as `w/o Gram` in Table 5. The average performance drops by $2.66\%$, confirming that Gram matrices are essential for effective knowledge transfer between modalities due to their information concentration properties.

**Second, text and vision modalities exhibit similar structural patterns in their Gram matrices**. Despite dimensional differences, we observe relatively stable eigenvalue distributions across indices for both modalities, indicating cross-modal structural similarities despite their dimensional differences. This structural similarity enables effective cross-modal knowledge transfer through the shared parameter space. To validate this, we use independent parameters instead of shared ones, denoted as `w/o Shared` in Table 5. The average performance drops by $2.78\%$, demonstrating that the emergent similar patterns are functionally critical for effective knowledge sharing.

**Third, we observe converging representational structures in deeper layers**. The eigenvalue pattern of Layer 11 shows more convergence compared to Layer 10, suggesting that deeper layers develop more aligned representational structures, which MoRA effectively leverages and maintains information preservation while enabling cross-modal transfer.

### 4.6 ABLATION STUDIES

**Compared to Parameter-Efficient Fine-Tuning Methods**   We compare other parameter-efficient fine-tuning techniques, including LoRA (Hu et al., 2022), DoRA (Liu et al., 2024a), and BitFit (Zaken et al., 2022), as shown in Table 4. Low-rank-based methods achieve the best performance due to their flexibility. MoRA outperforms other methods, demonstrating its effectiveness in enabling modality interaction.

**Alternatives for Addressing Dimension Mismatch**   As shown in Table 5, `Align` uses two extra linear layers to project modalities into the same embedding, while `Fusion` concatenates embed-

Table 6: Performance comparison on image-to-image retrieval using models trained on multimodal CIRR dataset. Results are evaluated on the MS-COCO validation set.

| Method | Recall@1 | Recall@5 | Recall@10 |
|---|---|---|---|
| CLIP4CIR (Baldrati et al., 2024) | 43.34 | 76.99 | 86.49 |
| CLIP4CIR + MoRA | **60.50** | **85.00** | **88.60** |

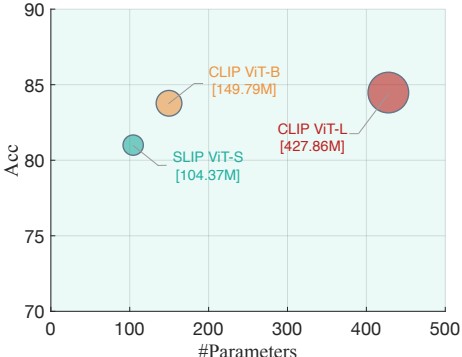

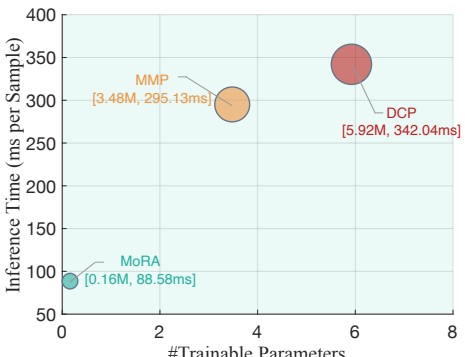

Figure 5: Performance scaling of MoRA with different backbone models.

Figure 6: Inference time (ms) per sample versus the number of trainable parameters.

dings from different modalities and uses one linear layer to project. MoRA consistently outperforms them across all datasets. We also conduct experiments removing modality-specific parameters in MoRA, denoted as w/o Specific. For Gram matrix construction, we report the performance of removing it and replacing it with a learnable one. These results validate the effectiveness of MoRA. We also add the ignored $\mathbf{W}_0$ in Equation (2), showing that the learnable magnitude parameters effectively compensate for the omitted frozen weights during training.

Parameter sensitivity analysis can be found in Section D.

### 4.7 EXTENSION TO EMBEDDING TASKS

To demonstrate MoRA's generalizability beyond classification tasks, we evaluate it on Composed Image Retrieval (CIR) based on Baldrati et al. (2024) using CLIP models, where models use a reference image and text modification to identify target images. CIR models are trained on multimodal inputs, making the original image-to-image retrieval as an important and natural missing-modality scenario where texts are absent. We train models on the CIRR dataset (Liu et al., 2021) with complete image-text pairs and evaluate on the MS-COCO validation set (Lin et al., 2014) for image-to-image retrieval, as shown in Table 6. MoRA achieves substantial improvements across all recall metrics, indicating that MoRA's applicability to tasks beyond classification, where missing modalities fundamentally alter the task dynamics. Implementation details can be found in Section A.3.

### 4.8 SCALABILITY AND INFERENCE TIME

Figure 5 demonstrates the effectiveness of MoRA integration across various backbone architectures, including SLIP ViT-S (Mu et al., 2022), CLIP ViT-B, and CLIP ViT-L (Radford et al., 2021). The results indicate that performance exhibits favorable scaling properties with respect to model capacity, with accuracy improvements correlating positively with the number of parameters. We conduct a comprehensive analysis of inference times to evaluate the computational efficiency of MoRA and prompt-based methods, including MMP and DCP. As shown in Figure 6, prompt-based methods significantly increase the inference time. MoRA theoretically introduces no inference overhead, and experimental results demonstrate its efficiency.

## 5 CONCLUSION

We introduced MoRA, a parameter-efficient fine-tuning method that effectively addresses the missing modality challenge in multimodal learning through shared cross-modal parameters and modality-specific adaptations. By leveraging Gram matrices for dimension-agnostic knowledge transfer, MoRA enables bidirectional information exchange while preserving modality-specific characteristics without introducing inference overhead. Extensive experiments demonstrate that MoRA significantly outperforms existing approaches across multiple benchmarks both on performance and inference time, demonstrating the effectiveness and efficiency of MoRA.

## 6 ACKNOWLEDGMENTS

This work was supported in part by Semiconductor Research Corporation JUMP 2.0 PRISM Center.

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
