# A DETAILS OF EXPERIMENTAL SETUP

## A.1 DATASET

We evaluate our proposed method on three standard benchmarks: MM-IMDb (Ovalle et al., 2017), UPMC-Food101 (Wang et al., 2015), and Hateful Memes (Kiela et al., 2020).

**MM-IMDb** represents the largest publicly available multimodal collection for movie genre prediction, containing 25,959 movies annotated with both visual and textual information. This dataset supports multi-label classification across 27 distinct movie genres. The corpus is structured with 15,552 training, 2,608 validation, and 7,799 test image-text pairs, providing a robust foundation for developing and evaluating multimodal classification models.

**UPMC Food101** is a comprehensive multimedia collection featuring noisy image-text pairs gathered from Google Image Search across 101 food categories. The dataset is structured with 61,127 training samples, 6,845 validation samples, and 22,716 test image-text pairs, providing substantial material for developing and evaluating multimodal food recognition systems.

**Hateful Memes** represents a benchmark multimodal collection for detecting hate speech in memes. It contains over 10,000 image-text pairs specifically designed to evaluate multimodal reasoning capabilities, where the interplay between text and visuals is crucial for accurate classification. The dataset consists of 8,500, 500, and 1,000 samples for training, validation, and testing.

## A.2 BASELINE METHODS

To evaluate MoRA, we select the SOTA missing-modality methods and multimodal prompt methods. Specifically, we select the missing modality methods, including MMP (Lee et al., 2023) and DCP (Hu et al., 2024); for multimodal prompt learning, we choose CoOp (Zhou et al., 2022), MaPLe (Khattak et al., 2023), and DePT (Zhang et al., 2024). Although a recent work, SyP (Zhang et al., 2025), employs the prompt-based method to address the missing modality task, the code for this work was not released upon our submission. Therefore, we do not compare our method with it. Once the source code or pre-trained models are released, we will add the results to the main results.

## A.3 IMPLEMENTATION DETAILS

**Main Experiments** Following previous work (Hu et al., 2024), we use CLIP ViT-B/16 as the backbone model. We add a fully connected layer at the top of the model as the classification layer for downstream tasks. The parameters in the CLIP model are frozen, and we only fine-tune the parameters of the classification layer and MoRA modules. MoRA can be inserted into various positions in the backbone. We found that the best hyper-parameters differ in various datasets. In UPMC-Food101, MoRA is inserted into the $\mathbf{Q}, \mathbf{V}$ in self-attention modules of the last two vision and text transformer layers. Rank $r$ is 4. We use the AdamW (Loshchilov & Hutter, 2019) optimizer with a learning rate $0.01$ and weight decay $0.02$. A linear warmup cosine annealing scheduler with $10\%$ warmup steps is used to adjust the learning rate. The batch size is $256$. The number of training epochs is $20$ and we apply the early-stopping strategy. Detailed settings can be found in the code we provided. If not specified, experiments are conducted on the UPMC-Food101 dataset with a missing ratio $\eta$ of $70\%$, where both image and text modalities are absent. We run experiments three times and report their average performance. All experiments are conducted on one NVIDIA H100 GPU.

**Motivation** In Figure 1, most hyper-parameters are the same as those used in the main experiments. We use CLIP ViT-B/16 as the aligned model and pre-trained ViT-B/16 and BERT as the unaligned model. We train these models on complete training datasets, i.e., $\eta = 0\%$, and test them using different datasets, including complete, image, and text-only datasets.

**Embedding Task** In Table 6, we use the same settings in the main experiments above. We use the CIRR training data as the training set, and evaluate the trained model on the MS-COCO validation set. We select CLIP ViT-B/16 as the backbone. For training, we use the complete samples without any modality-incomplete data. During testing, the model is evaluated on the image-to-image retrieval task, which can be viewed as if the text modality is missing.

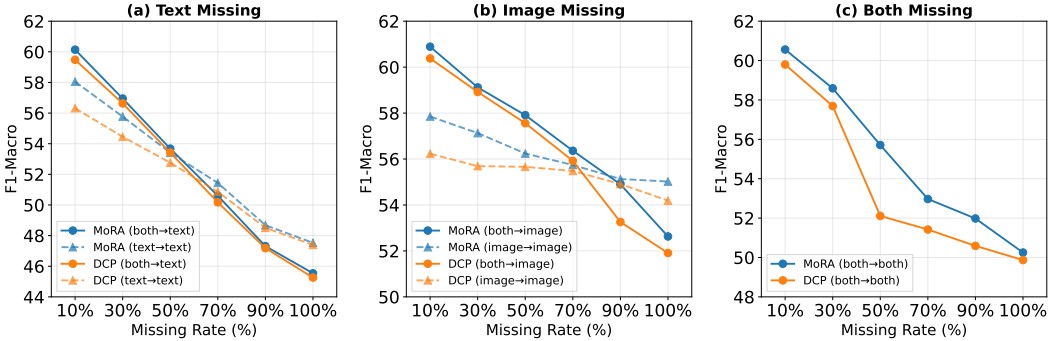

Figure 7: Performance comparison on MM-IMDb with varying missing ratios.

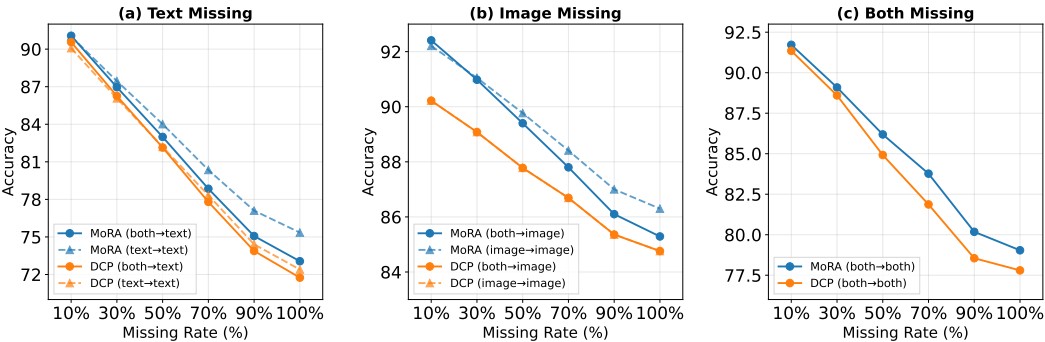

Figure 8: Performance comparison on Food101 with varying missing ratios.

## B ATTACHED POSITION

We systematically evaluate MoRA attachment at different network depths, as shown in Figure 9. To further analyze the effect of attached positions, we attached MoRA to three positions of a CLIP ViT-B/16 model:

- Front layers: Layers 1-2 (early feature extraction)

- Middle layers: Layers 6-7 (intermediate representations)

- Rear layers: Layers 11-12 (high-level semantics)

We train the model on the Food101 dataset with a $70\%$ missing ratio and both modalities are missing. As shown in Table 7, attaching to deeper layers enables fine-tuning of high-level semantic features rather than low-level representations, yielding superior performance. This strategy effectively handles architectures with asymmetric depths. As demonstrated in Figure 5, MoRA successfully adapts CLIP ViT-L (24 vision layers, 12 text layers) by consistently targeting the final layers of each modality, which contain the most semantic information.

## C MORE RESULTS

More results across various missing ratios are shown in Figure 7 and Figure 8. The experimental results demonstrate consistent effectiveness in handling missing modalities. As the missing ratio increases, performance on all datasets gradually declines. Notably, the text-only modality consistently outperformed image-only across all datasets. MoRA maintains robust performance even at high missing ratios, preserving inter-modality interactions while maintaining intra-modality flexibility.

Table 7: Performance comparison of attached positions.

| Position | Front | Middle | **Rear (Ours)** |
|---|---|---|---|
| Accuracy | 81.08 | 82.48 | **83.77** |

Figure 9: Parameter sensitivity analysis.

## D PARAMETER SENSITIVITY

The sensitivity of MoRA to its key hyper-parameters is shown in Figure 9, where $r$ denotes the rank, $\eta$ is the low-rank strength, "#Layers" denotes the number of layers inserted by MoRA, and "Position" means which attention matrices are adjusted. The results show that MoRA is robust to parameter changes, maintaining strong performance across a wide range of values.

## E VISUALIZATION OF EMBEDDING SPACE

To further analyze why MoRA outperforms other methods, we use t-SNE (Van der Maaten & Hinton, 2008) to visualize the embeddings of samples with missing modalities, as illustrated in Figure 10. Specifically, we use the samples in the test dataset, obtain the embeddings from available modalities, and visualize them. The results show that the embedding space of FFT has collapsed, and MoRA produces more compact and well-separated clusters. Compared to DCP, MoRA has a larger inter-class distance, indicating better discriminability.

## F LLM USAGE STATEMENT

Large language models were used as a general-purpose writing assistance tool during the preparation of this manuscript, primarily for grammar checking, sentence restructuring, and improving clarity of technical descriptions. LLMs did not contribute to the core research ideas, experimental design, or technical innovations presented in this work. All scientific claims, experimental results, and theoretical contributions are the original work of the authors, who take full responsibility for the accuracy and integrity of all content.

## G LIMITATIONS

While MoRA demonstrates strong performance across various missing-modality scenarios, several limitations present opportunities for future research.

First, our experimental validation is limited to three datasets (MM-IMDb, UPMC-Food101, and Hateful Memes) and primarily focuses on image-text modality pairs. Future work could extend MoRA to additional multimodal domains (e.g., audio-visual) and more diverse datasets to further validate its generalizability. Second, the current formulation of MoRA addresses binary missing-modality scenarios (present or absent). Future work could explore extensions to partial or corrupted

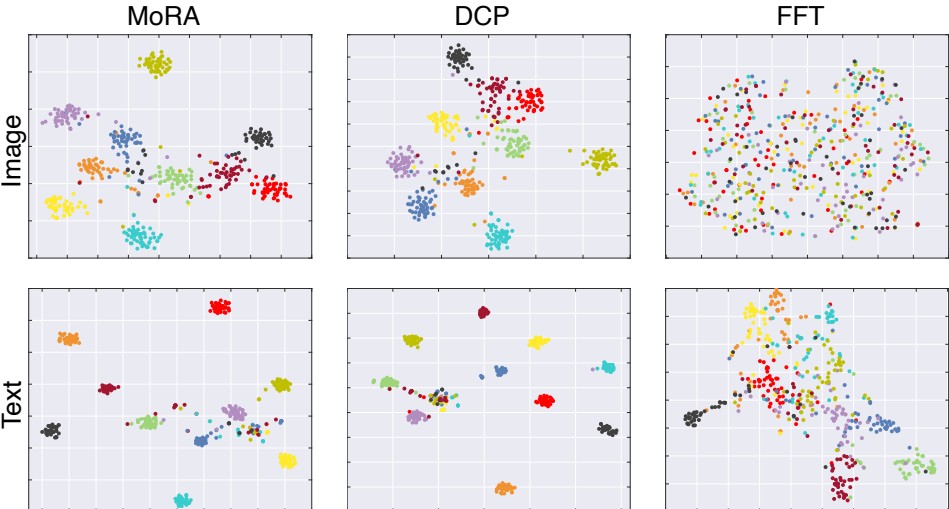

Figure 10: t-SNE visualization for MoRA, DCP, and FFT. Different colors denote different categories.

modalities where information is present but degraded, which may better reflect certain real-world applications.

Despite these limitations, MoRA represents a significant step forward in addressing the missing modality challenge through its novel parameter-efficient fine-tuning approach.

## H  BROADER IMPACTS

Our research on MoRA offers several positive societal impacts. By addressing the missing modality challenge in multimodal systems, MoRA can significantly improve accessibility for users with sensory impairments who may not have access to all modalities. Additionally, MoRA reduces computational requirements and potentially lowers energy consumption compared to alternative methods, contributing to more sustainable AI development. This efficiency also enables more robust deployment of multimodal systems in resource-constrained environments like healthcare, education, and humanitarian assistance. Furthermore, MoRA could enhance privacy by allowing users to selectively withhold certain modalities while still receiving reasonable system performance.

We also acknowledge potential concerns regarding this technology. As with many AI advancements, improvements in handling missing modalities could potentially be applied in ways that raise privacy questions if deployed without appropriate safeguards. Additionally, systems making decisions based on incomplete information should be deployed with appropriate human oversight, particularly in high-stakes applications. We've focused our development on public benchmark datasets and emphasize that our primary goal is improving the accessibility, efficiency, and robustness of multimodal systems rather than enabling capabilities that could raise significant ethical concerns.