# OpenReview forum: "MoRA: Missing Modality Low-Rank Adaptation for Visual Recognition"
_ICLR.cc/2026/Conference — ICLR 2026 Poster_

### Official Review · Reviewer_AxMw · 2025-10-28

**Soundness:** 4
**Presentation:** 3
**Contribution:** 3
**Rating:** 6
**Confidence:** 4

**Summary:**

This paper introduces MoRA (Missing Modality Low-Rank Adaptation), a parameter-efficient fine-tuning method for vision-language models (VLMs) under missing modality scenarios.

MoRA addresses this by introducing two types of parameters:
* Modality-specific parameters for independent adaptation of each modality.
* Shared cross-modal parameters that facilitate bidirectional knowledge transfer via Gram matrices in a low-rank space.

This allows MoRA to maintain inter-modality alignment while preserving intra-modality flexibility, without introducing inference overhead.

**Strengths:**

Novel contribution: The use of Gram matrices for cross-modal low-rank adaptation is original and mathematically elegant, avoiding dimension mismatch between modalities.

Comprehensive evaluation: The paper evaluates across multiple datasets, missing ratios (50–90%), and cross-scenario generalization settings.

Strong empirical results: MoRA outperforms baselines in accuracy.

Efficiency: Achieves high performance with minimal trainable parameters and no inference overhead, a key improvement over prompt-based tuning.

Insightful analyses: Includes embedding-space, eigenvalue, and ablation studies supporting claims of inter-modal alignment and modality-specific adaptability.

Extensible framework: Demonstrates that MoRA generalizes beyond classification to retrieval tasks (CIRR→COCO).

**Weaknesses:**

Evaluation diversity: All experiments use CLIP ViT-B/16 as the backbone; results on other architectures are only briefly summarized (Fig. 5).

Limited modality scope: Experiments are restricted to image–text tasks. The paper claims generality but does not test on other modality pairs.

Hyperparameters were likely tuned on the test set and differ from dataset to dataset. Hyperparameters should be either selected on a validation set or on one dataset and shared across all datasets.

Experiments are restricted to image-text tasks, despite claiming generality, the method is not tested on audio–visual or video–text datasets.

**Questions:**

Could the Gram matrix mechanism be extended to more than two modalities (e.g., vision, text, and audio)?

How well does MoRa perform when it uses the same hyperparameters for all datasets (for example, the one for Food101)

How sensitive is MoRA on the rank?

Does MoRA’s shared Gram-space adaptation still yield improvements when applied to unaligned encoders?

---

> ### Author Response · Authors · 2025-11-20
>
> We sincerely thank the reviewer for the thoughtful and constructive feedback. We greatly appreciate your recognition of our "_novel contribution_", the "_strong empirical results_," and our method's "_efficiency_". We address each of your valuable questions in detail below.
>
> ---
>
> >***`Q1`: Extend to single-branch models.***
>
> A: Thank you for this valuable suggestion. We conduct comprehensive experiments demonstrating that **MoRA generalizes effectively to single-branch architectures**.
>
> **Experimental Setup**
> * **Backbone:** `ViLT` [1], a representative single-branch (early-fusion) model. The reason we chose it is that it is the backbone model used in MMP [2], enabling us to compare against established baselines for fair evaluation.
> * **Datasets:** `MM-IMDb`, `UPMC-Food101`, `Hateful Memes`
> * **Missing Settings:** `70%` missing ratio, `both` modalities missing (train & test).
> * **Hyperparameters:** We use the same hyperparameters reported in the main paper.
> * **Implementation Details:** MoRA is a general framework that receives information from different modalities, fuses them, and sends them back to the original transformer blocks during training. We identify vision and text tokens using the indices of modality-type embeddings within a single transformer layer as inputs.
>
> **Results**
>
> |               | MMP   | DCP   | MoRA (Ours) |
> | ------------- | ----- | ----- | ----------- |
> | MM-IMDb       | 42.66 | 48.45 | **48.47**   |
> | UPMC-Food101  | 79.08 | 80.85 | **80.91**   |
> | Hateful Memes | 66.07 | 66.68 | **68.17**   |
>
> **Key Insights**
> * **Architectural Agnostic:** MoRA successfully adapts to single-branch architectures and maintains the performance compared to prompt-based baselines.
> * **Zero Inference Overhead:** MoRA introduces no overhead during inference.
>
>  >***`Q2 & Q4 & Q5`: Extend to modalities beyond vision and text, and more modalities***
>
> A: Thank you for this important question. We conduct experiments demonstrating that **MoRA scales effectively beyond vision-language to diverse modality combinations**.
>
> **Experimental Setup**
> * **Backbone:** `ImageBind` [3], a unified model supporting six modalities (images, text, audio, depth, thermal, IMU), chosen to evaluate MoRA's generalization beyond image-text modalities.
> * **Dataset:** `RGB-D Scenes Dataset v2` [4] with 14 scene categories
> * **Data Split:** `1,828` training samples (80%) / `463` validation samples (20%)
> * **Missing Settings:** `Both` modalities (RGB+Depth) missing with varying ratios (`50%`, `70%`, `90%`)
> * **Hyperparameters:** Same as reported in the main paper for both MoRA and DCP.
>
> **Results**
>
> | Missing Ratio | DCP   | MoRA (Ours) |
> | ------------- | ----- | ----------- |
> | 50%           | 85.10 | **89.20**   |
> | 70%           | 82.07 | **87.90**   |
> | 90%           | 80.35 | **86.39**   |
>
> **Key Insights**
> * **Beyond Vision-Language:** MoRA successfully handles RGB-Depth fusion, confirming its applicability beyond vision and text.
> * **Consistent Gains:** Performance improvements highlight MoRA's effectiveness.
>
> We also provide experiments demonstrating that **MoRA can be extended to more than 2 modalities.**
>
> **Experimental Setup**
> * **Backbone:** `ImageBind` [3], a unified model supporting six modalities (images, text, audio, depth, thermal, IMU), chosen to evaluate MoRA's generalization beyond image-text modalities.
> * **Dataset:** `RGB-D Scenes Dataset v2` [4] with 14 scene categories
> * **Data Split:** `1,828` training samples (80%) / `463` validation samples (20%)
> * **Modalities**
> 	* Original: `RGB`, `Depth`
> 	* Curation: `Text` descriptions (generated via GPT-4o-mini from RGB images without label leakage)
> * **Text Generation Prompt:** "Describe this indoor scene image in detail. Focus on visible objects, furniture, layout, and spatial characteristics. DO NOT mention or reference that this is a '{scene_name}'."
> * **Missing Settings:** `70%` missing ratio with (1) `single` modality missing, (2) `two` modalities missing
> * **Hyperparameters:** Same as reported in the main paper for both MoRA and DCP.
>
> **Results**
>
> |                        | DCP   | MoRA (Ours) |
> | ---------------------- | ----- | ----------- |
> | Missing One Modality   | 80.46 | **91.01**   |
> | Missing Two Modalities | 75.89 | **84.83**   |
>
> **Key Insights**
> * **Robustness and Linear Scalability.** MoRA can be extended to more than two modalities and demonstrates its performance.

---

> ### Author Response · Authors · 2025-11-20
>
> >***`Q3 & Q6 & Q7`: Hyperparameter Robustness***
>
> A: Thank you for this question. We conduct comprehensive ablation studies on **MoRA's architectural placement** across MM-IMDb and Hateful Memes datasets to complement the UPMC-Food101 analysis in Figure 9 (Appendix).
>
> **Experimental Setup**
> * **Backbone:** `CLIP-ViT-B/16`
> * **Datasets:** `MM-IMDb`, `Hateful Memes`
> * **Missing Settings:** `70%` missing ratio, `both` modalities missing
> * **Hyperparameters:** Same as main paper
>
> **Results**
>
> | MM-IMDb | q     | k     | v     | qk    | qv    | kv    | qkv   |
> | ------- | ----- | ----- | ----- | ----- | ----- | ----- | ----- |
> | 8       | 53.07 | 53.41 | 52.27 | 51.58 | 52.65 | 52.67 | 50.85 |
> | 9       | 53.03 | 52.63 | 52.83 | 52.97 | 52.44 | 53.04 | 51.92 |
> | 10      | 52.00 | 52.08 | 52.13 | 52.59 | 52.97 | 52.72 | 52.49 |
> | 11      | 52.54 | 52.39 | 53.37 | 51.07 | 52.58 | 51.80 | 52.44 |
>
> | Hateful Memes | q     | k     | v     | qk    | qv    | kv    | qkv   |
> | ------------- | ----- | ----- | ----- | ----- | ----- | ----- | ----- |
> | 8             | 64.37 | 66.76 | 66.88 | 62.25 | 67.14 | 64.04 | 65.47 |
> | 9             | 65.72 | 67.88 | 67.61 | 66.75 | 67.96 | 66.18 | 67.22 |
> | 10            | 66.03 | 66.73 | 68.64 | 62.67 | 70.15 | 69.69 | 68.53 |
> | 11            | 66.31 | 68.32 | 68.10 | 67.68 | 68.25 | 68.70  | 68.96 |
>
> **Key Insights**
> * **Architectural Robustness:** Performance variation across all configurations is minimal, confirming MoRA's stability.
>
> **The performance regarding Rank $r$ has been reported in Figure 9, Appendix.** We report the performance on other datasets using the same settings.
>
> **Experimental Setup**
> * **Backbone:** `CLIP-ViT-B/16`
> * **Datasets:** `MM-IMDb`, `Hateful Memes`
> * **Missing Settings:** `70%` missing ratio, `both` modalities missing
> * **Hyperparameters:** Same as main paper
>
> **Results**
>
> |               | 1     | 2     | 4     | 8     | 16    | 32    | 64    |
> | ------------- | ----- | ----- | ----- | ----- | ----- | ----- | ----- |
> | MM-IMDb       | 51.03 | 52.14 | 52.97 | 53.01 | 51.67 | 52.67 | 52.89 |
> | Hateful Memes | 69.52 | 69.03 | 70.01 | 70.15 | 70.08 | 69.97 | 70.13 |
>
> **Key Insights**
> * **Robustness of hyperparameters.** MoRA is robust to different hyperparameters.
>
> ---
>
> >***`Q8`: Extent MoRA to Unaligned Encoders.***
>
> A: Good question! We report the performance using the same settings with `Figure 1` and `Section A.3 (Appendix)`.
>
> **Experimental Setup**
> * **Backbone:** `vit-base-patch16-224` (image) and `bert-base-uncased` (text)
> * **Datasets:** `UPMC-Food101`
> * **Missing Settings:** `0%` training missing ratio. `0%` testing missing ratio + `100%` testing missing `image` + `100%` testing missing `text`.
> * **Hyperparameters:** Same as main paper
>
> **Results**
>
> |          | 0%    | 100% Image | 100% Text |
> | -------- | ----- | ---------- | --------- |
> | w/ MoRA  | **84.50** | **27.19**      | **62.01**     |
> | w/o MoRA | 77.70 | 1.20       | 1.40      |
>
> **Key Insights**
> * **Extensibility of MoRA.** MoRA can be integrated with unaligned encoders to facilitate cross-modal interactions to improve the performance.
>
> ---
>
> **Reference**
>
> [1] Kim, Wonjae, Bokyung Son, and Ildoo Kim. "Vilt: Vision-and-language transformer without convolution or region supervision." ICML 2021.
>
> [2] Lee, Yi-Lun, et al. "Multimodal prompting with missing modalities for visual recognition." CVPR 2023.
>
> [3] Girdhar, Rohit, et al. "Imagebind: One embedding space to bind them all." CVPR 2023.
>
> [4] Lai, Kevin, Liefeng Bo, and Dieter Fox. "Unsupervised feature learning for 3d scene labeling." ICRA 2014.

---

> > ### Comment · Reviewer_AxMw · 2025-11-25
> >
> > The concerns and questions I raised in my initial review have been addressed and answered. Thank you for the clarifications.

---

> ### Author Response · Authors · 2025-11-25
>
> Dear Reviewer `AxMw`,
>
> We're glad to hear that our response helped address your concerns. Thank you again for your recognition and support!
>
> Best regards,
>
> The Authors of Submission 20667

---

### Official Review · Reviewer_3Yzm · 2025-11-01

**Soundness:** 4
**Presentation:** 3
**Contribution:** 3
**Rating:** 6
**Confidence:** 4

**Summary:**

This paper addresses the missing modality problem in visual recognition tasks and proposes missing modality low-rank adaptation (MoRA), a parameter-efficient fine-tuning approach that explicitly models cross-modal interactions while maintaining modality-specific adaptations. Specifically, the MoRA introduces shared cross-modal parameters based on the Gram Matrix to enable bidirectional knowledge transfer, along with modality-specific adapting parameters to preserve the unique characteristics of each modality. Extensive experiments and ablation studies are conducted to demonstrate the efficacy of the proposed method.

**Strengths:**

+ The paper is technically sound, well-motivated, and tackles the missing modality issue, which is an important and practical challenge in multimodal learning.
+ The proposed MoRA method is conceptually interesting and carefully designed, as it explicitly models cross-modal interactions while retaining modality-specific adaptations.
+ The proposed MoRA brings impressive improvements and consistently outperforms other SOTA methods across various missing-modality scenarios.
+ Extensive ablation studies are conducted to validate the contribution of each component and showcase the efficiency of MoRA in both training and inference time and computational cost.

**Weaknesses:**

- Scalability to more modalities (e.g., beyond two) is a potential concern. While MoRA appears feasible for dual-modality tasks such as image–text or audio–text learning, its current design may not scale efficiently to scenarios involving multiple modalities (e.g., 4 or more), where cross-modal adaptations could grow exponentially in cost and complexity.
- The generalizability across multimodal architectures is not clearly demonstrated. The proposed MoRA is primarily evaluated on a CLIP-like architecture, which has a separate encoder for each modality. However, for architectures such as ViLT or InstructBLIP, where early fusion occurs and modality interactions are embedded within transformer attention layers, it remains unclear whether the proposed adaptation strategy would still be directly applicable or equally effective.

**Questions:**

Please address my concerns in the weaknesses section.

---

> ### Author Response · Authors · 2025-11-20
>
> We sincerely thank the reviewer for the thoughtful and constructive feedback. We greatly appreciate your recognition of our method as "_technically sound and well-motivated_," "_conceptually interesting and carefully designed_," and bringing "_impressive improvements_." We address each of your valuable questions in detail below.
>
> ---
>
> > ***`Q1`: Scalability to more modalities.***
>
> A: Good question! MoRA consists of modality-specific and modalities-shared modules (`Eq 4`), and **scales linearly O(N) with the number of modalities.** Specifically, considering the third modality depth $\mathbf{S}^{\mathrm{d}}$ and corresponding Gram matrix $\mathbf{G}^{\mathrm{d}}$,  the updated weights $\Delta \mathbf{W}^{\mathrm{v}} = {\mathbf{S}^{\mathrm{v}}}^T\mathbf{G}^{\mathrm{t}}\mathbf{G}^{\mathrm{d}}\mathbf{S}^{\mathrm{v}}$. To explore the effectiveness under this scenario, we conduct experiments using the following settings.
>
> **Experimental Setup**
> * **Backbone:** `ImageBind` [1], a unified model supporting six modalities (images, text, audio, depth, thermal, IMU), chosen to evaluate MoRA's generalization beyond image-text modalities.
> * **Dataset:** `RGB-D Scenes Dataset v2` [2] with 14 scene categories
> * **Data Split:** `1,828` training samples (80%) / `463` validation samples (20%)
> * **Modalities**
> 	* Original: `RGB`, `Depth`
> 	* Curation: `Text` descriptions (generated via GPT-4o-mini from RGB images without label leakage)
> * **Text Generation Prompt:** "Describe this indoor scene image in detail. Focus on visible objects, furniture, layout, and spatial characteristics. DO NOT mention or reference that this is a '{scene_name}'."
> * **Missing Settings:** `70%` missing ratio with (1) `single` modality missing, (2) `two` modalities missing
> * **Hyperparameters:** Same as reported in the main paper for both MoRA and DCP.
>
> **Results**
>
> |                        | DCP   | MoRA (Ours) |
> | ---------------------- | ----- | ----------- |
> | Missing One Modality   | 80.46 | **91.01**   |
> | Missing Two Modalities | 75.89 | **84.83**   |
>
> **Key Insights**
> * **Robustness and Linear Scalability.** MoRA can be extended to more than two modalities and demonstrates its performance.
>
> ---
>
>  >***`Q2`: Extend to single-branch models.***
>
> A: Thank you for this valuable suggestion. We conduct comprehensive experiments demonstrating that **MoRA generalizes effectively to single-branch architectures**.
>
> **Experimental Setup**
> * **Backbone:** `ViLT` [3], a representative single-branch (early-fusion) model. The reason we chose it is that it is the backbone model used in MMP [4], enabling us to compare against established baselines for fair evaluation.
> * **Datasets:** `MM-IMDb`, `UPMC-Food101`, `Hateful Memes`
> * **Missing Settings:** `70%` missing ratio, `both` modalities missing (train & test).
> * **Hyperparameters:** We use the same hyperparameters reported in the main paper.
> * **Implementation Details:** MoRA is a general framework that receives information from different modalities, fuses them, and sends them back to the original transformer blocks during training. We identify vision and text tokens using the indices of modality-type embeddings within a single transformer layer as inputs.
>
> **Results**
>
> |               | MMP   | DCP   | MoRA (Ours) |
> | ------------- | ----- | ----- | ----------- |
> | MM-IMDb       | 42.66 | 48.45 | **48.47**   |
> | UPMC-Food101  | 79.08 | 80.85 | **80.91**   |
> | Hateful Memes | 66.07 | 66.68 | **68.17**   |
>
> **Key Insights**
> * **Architectural Agnostic:** MoRA successfully adapts to single-branch architectures and maintains the performance compared to prompt-based baselines.
> * **Zero Inference Overhead:** MoRA introduces no overhead during inference.
>
> ---
>
> **Reference**
>
> [1] Girdhar, Rohit, et al. "Imagebind: One embedding space to bind them all." CVPR 2023.
>
> [2] Lai, Kevin, Liefeng Bo, and Dieter Fox. "Unsupervised feature learning for 3d scene labeling." ICRA 2014.
>
> [3] Kim, Wonjae, Bokyung Son, and Ildoo Kim. "Vilt: Vision-and-language transformer without convolution or region supervision." ICML 2021.
>
> [4] Lee, Yi-Lun, et al. "Multimodal prompting with missing modalities for visual recognition." CVPR 2023.

---

> ### Author Response · Authors · 2025-11-27
>
> We believe that our follow-up experiment results and explanations address the concerns raised by the reviewer. Since the rebuttal window will close within one week, we respectfully ask that they consider raising their score to reflect these clarifications and improvements if the reviewer finds them satisfactory. We are happy to answer any remaining questions.

---

### Official Review · Reviewer_fEg2 · 2025-11-01

**Soundness:** 2
**Presentation:** 3
**Contribution:** 2
**Rating:** 4
**Confidence:** 4

**Summary:**

This paper addresses performance degradation in Vision-Language Models (VLMs) when a modality is missing, a scenario where existing prompt-based solutions suffer from high inference latency. The authors propose MoRA (Missing Modality Low-Rank Adaptation), a new Parameter-Efficient Fine-Tuning (PEFT) method that combines modality-specific adapters with novel shared low-rank parameters to model cross-modal interactions. Experiments on three benchmarks show MoRA significantly outperforms prior SOTA methods like DCP in missing-modality scenarios while being substantially more efficient at inference time.

**Strengths:**

- The core mechanism of MoRA is clever. Using Gram matrices to facilitate shared, low-rank adaptation between two frozen encoders of different dimensions is a technical solution to the dimension-mismatch problem.
- The primary advantage of MoRA over its main competitors (MMP, DCP) is its efficiency. Because MoRA is a LoRA-based method, all its adapter weights can be merged into the backbone model post-training.

**Weaknesses:**

- The central premise of the paper—that missing modalities are a common, critical "real-world scenario"—is weakly motivated. The paper justifies this by vaguely citing "privacy constraints, collection difficulties, or resource limitations" but provides no concrete, compelling examples.
- In most practical VLM applications (VQA, captioning, retrieval), the user provides the modalities. The experimental scenarios, such as classifying a "hateful meme" with the text missing, or identifying food from a recipe with the image missing, feel highly contrived and niche.
- The proposed solution is architecturally specific to dual-encoder models like CLIP. It is not clear how this Gram-matrix-based cross-encoder sharing would be applied to modern, single-stack, fused MLLMs (e.g., LLaVA-style models), which are becoming the standard.

**Questions:**

Please see weaknesses.

---

> ### Author Response · Authors · 2025-11-20
>
> We sincerely thank the reviewer for the thoughtful and constructive feedback. We greatly appreciate your recognition of our "*Innovation of proposed method*" and "*Inference time efficiency*." We address each of your valuable questions in detail below.
>
> ---
>
> > ***`Q1 & Q2`: Missing Modality Applications.***
>
> A: Thanks for pointing out this question. Although multimodal large language models have achieved significant progress in recent years, they are not the only topic in the computer vision/multimodality research community, especially for **edge devices** and intersection with other areas like **medicine**. We kindly refer [1] to you, which introduces the applications, including:
> * A **social network** might be unable to access location information if users decline to share their private location.
> * A **healthcare application** might not have all the records available when patients are unwilling to undergo risky or invasive examinations.
>
> We provide an extra example regarding **Autonomous Vehicles**:
> * Due to the incident, one of the sensors may be disabled, and the vehicle must be robust to handle the emergent scenario [2].
>
> MoRA is a generic framework that supports various downstream tasks, as reported in Tables 1 and 6. We conduct additional experiments showing that MoRA can also be integrated with single-branch architectures and with additional modalities (see Q3 response).
>
> ---
>
> > ***`Q3`: Extend to single-branch models.***
>
> A: Thank you for this valuable suggestion. We conduct comprehensive experiments demonstrating that **MoRA generalizes effectively to single-branch architectures**.
>
> **Experimental Setup**
> * **Backbone:** `ViLT` [3], a representative single-branch (early-fusion) model. The reason we chose it is that it is the backbone model used in MMP [4], enabling us to compare against established baselines for fair evaluation.
> * **Datasets:** `MM-IMDb`, `UPMC-Food101`, `Hateful Memes`
> * **Missing Settings:** `70%` missing ratio, `both` modalities missing (train & test).
> * **Hyperparameters:** We use the same hyperparameters reported in the main paper.
> * **Implementation Details:** MoRA is a general framework that receives information from different modalities, fuses them, and sends them back to the original transformer blocks during training. We identify vision and text tokens using the indices of modality-type embeddings within a single transformer layer as inputs.
>
> **Results**
>
> |               | MMP   | DCP   | MoRA (Ours) |
> | ------------- | ----- | ----- | ----------- |
> | MM-IMDb       | 42.66 | 48.45 | **48.47**   |
> | UPMC-Food101  | 79.08 | 80.85 | **80.91**   |
> | Hateful Memes | 66.07 | 66.68 | **68.17**   |
>
> **Key Insights**
> * **Architectural Agnostic:** MoRA successfully adapts to single-branch architectures and maintains the performance compared to prompt-based baselines.
> * **Zero Inference Overhead:** MoRA introduces no overhead during inference.
>
> We also provide experiments demonstrating that **MoRA scales effectively beyond vision-language to diverse modality combinations.**
>
> **Experimental Setup**
> * **Backbone:** `ImageBind` [5], a unified model supporting six modalities (images, text, audio, depth, thermal, IMU), chosen to evaluate MoRA's generalization beyond image-text modalities.
> * **Dataset:** `RGB-D Scenes Dataset v2` [6] with 14 scene categories
> * **Data Split:** `1,828` training samples (80%) / `463` validation samples (20%)
> * **Missing Settings:** `Both` modalities (RGB+Depth) missing with varying ratios (`50%`, `70%`, `90%`)
> * **Hyperparameters:** Same as reported in the main paper for both MoRA and DCP.
>
> **Results**
>
> | Missing Ratio | DCP   | MoRA (Ours) |
> | ------------- | ----- | ----------- |
> | 50%           | 85.10 | **89.20**   |
> | 70%           | 82.07 | **87.90**   |
> | 90%           | 80.35 | **86.39**   |
>
> **Key Insights**
> * **Beyond Vision-Language:** MoRA successfully handles RGB-Depth fusion, confirming its applicability beyond vision and text.
> * **Consistent Gains:** Performance improvements highlight MoRA's effectiveness.
>
> ---
>
> **Reference**
>
> [1] Ma, Mengmeng, et al. "Are multimodal transformers robust to missing modality?." CVPR 2022.
>
> [2] Hao, Xiaoshuai, et al. "SafeMap: Robust HD Map Construction from Incomplete Observations." ICML 2025.
>
> [3] Kim, Wonjae, Bokyung Son, and Ildoo Kim. "Vilt: Vision-and-language transformer without convolution or region supervision." ICML 2021.
>
> [4] Lee, Yi-Lun, et al. "Multimodal prompting with missing modalities for visual recognition." CVPR 2023.
>
> [5] Girdhar, Rohit, et al. "Imagebind: One embedding space to bind them all." CVPR 2023.
>
> [6] Lai, Kevin, Liefeng Bo, and Dieter Fox. "Unsupervised feature learning for 3d scene labeling." ICRA 2014.

---

> ### Author Response · Authors · 2025-11-27
>
> We believe that our follow-up experiment results and explanations address the concerns raised by the reviewer. Since the rebuttal window will close within one week, we respectfully ask that they consider raising their score to reflect these clarifications and improvements if the reviewer finds them satisfactory. We are happy to answer any remaining questions.

---

### Official Review · Reviewer_qZrT · 2025-11-02

**Soundness:** 3
**Presentation:** 3
**Contribution:** 2
**Rating:** 6
**Confidence:** 4

**Summary:**

This paper presents MoRA (Missing Modality Low-Rank Adaptation), an innovative approach for parameter-efficient fine-tuning (PEFT) that tackles the issue of absent modalities in pre-trained Vision-Language Models (VLMs) during both training and inference. The proposed approach introduces modality specific and shared parameters with minimal computational overhead. The proposed methodology demonstrates superior performance compared to baseline models across multiple benchmarks involving missing modalities.

**Strengths:**

* Inclusion of low rank updates for modality specific and shared parameters.
* Inference time efficiency in terms of minimal computational overhead.
* Superior performance across multiple benchmarks involving diverse missing modality scenarios.
* Strong cross-scenario generalization in cases when training time missing-modality patterns differ significantly from the testing patterns.

**Weaknesses:**

* The proposed low rank update approach is restricted to the dual encoder models i.e. CLIP or SLIP. The applicability of the proposed approach should be explored for single stream models that relies on alignment of visual representations with LLM inputs.
* The current set of analysis is restricted to image-text datasets with significant text modality dominance. Tasks involving other modality combinations ( audio-visual inputs ) can be considered e.g. audio-visual action recognition.
* Situations involving partial modality corruptions are not considered. Examples include image corruptions by black patches and text corruption by random word dropouts or redaction by [MASK] tokens.
* Detailed analysis regarding the layer placement of MoRA updates w.r.t. individual datasets (MM-IMDb and Hateful-Memes) are not mentioned.

**Questions:**

* What is the sensitivity of proposed method MoRA to the initialization of the matrices Sv and St ?
* Further, have the authors explored any constraints on the shared parameters e.g. effect of orthonormal columns etc ?
* Have the authors considered a scaling factor between the modality specific and shared modality based updates (Eq 2 and 4) ?

---

> ### Author Response · Authors · 2025-11-20
> **Rebuttal by Authors**
>
> We sincerely thank the reviewer for the thoughtful and constructive feedback. We greatly appreciate your recognition of our "*Inference time efficiency*," "*Superior performance*," and "*Strong cross-scenario generalization*." We address each of your valuable questions in detail below.
>
> ---
>
> > ***`Q1`: Extend to single-branch models.***
>
> A: Thank you for this valuable suggestion. We conduct comprehensive experiments demonstrating that **MoRA generalizes effectively to single-branch architectures**.
>
> **Experimental Setup**
> * **Backbone:** `ViLT` [1], a representative single-branch (early-fusion) model. The reason we chose it is that it is the backbone model used in MMP [2], enabling us to compare against established baselines for fair evaluation.
> * **Datasets:** `MM-IMDb`, `UPMC-Food101`, `Hateful Memes`
> * **Missing Settings:** `70%` missing ratio, `both` modalities missing (train & test).
> * **Hyperparameters:** We use the same hyperparameters reported in the main paper.
> * **Implementation Details:** MoRA is a general framework that receives information from different modalities, fuses them, and sends them back to the original transformer blocks during training. We identify vision and text tokens using the indices of modality-type embeddings within a single transformer layer as inputs.
>
> **Results**
>
> |               | MMP   | DCP   | MoRA (Ours) |
> | ------------- | ----- | ----- | ----------- |
> | MM-IMDb       | 42.66 | 48.45 | **48.47**   |
> | UPMC-Food101  | 79.08 | 80.85 | **80.91**   |
> | Hateful Memes | 66.07 | 66.68 | **68.17**   |
>
> **Key Insights**
> * **Architectural Agnostic:** MoRA successfully adapts to single-branch architectures and maintains the performance compared to prompt-based baselines.
> * **Zero Inference Overhead:** MoRA introduces no overhead during inference.
>
> ---
>
> > ***`Q2`: Extend to modalities beyond vision and text.***
>
> A: Thank you for this important question. We conduct experiments demonstrating that **MoRA scales effectively beyond vision-language to diverse modality combinations**.
>
> **Experimental Setup**
> * **Backbone:** `ImageBind` [3], a unified model supporting six modalities (images, text, audio, depth, thermal, IMU), chosen to evaluate MoRA's generalization beyond image-text modalities.
> * **Dataset:** `RGB-D Scenes Dataset v2` [4] with 14 scene categories
> * **Data Split:** `1,828` training samples (80%) / `463` validation samples (20%)
> * **Missing Settings:** `Both` modalities (RGB+Depth) missing with varying ratios (`50%`, `70%`, `90%`)
> * **Hyperparameters:** Same as reported in the main paper for both MoRA and DCP.
>
> **Results**
>
> | Missing Ratio | DCP   | MoRA (Ours) |
> | ------------- | ----- | ----------- |
> | 50%           | 85.10 | **89.20**   |
> | 70%           | 82.07 | **87.90**   |
> | 90%           | 80.35 | **86.39**   |
>
> **Key Insights**
> * **Beyond Vision-Language:** MoRA successfully handles RGB-Depth fusion, confirming its applicability beyond vision and text.
> * **Consistent Gains:** Performance improvements highlight MoRA's effectiveness.
>
> ---
>
> > ***`Q3`: Extend to corrupted datasets.***
>
> A: Thank you for this insightful suggestion. We conduct comprehensive experiments demonstrating that **MoRA maintains robust performance under various modality corruption scenarios**.
>
> **Experimental Setup**
> * **Backbone:** `CLIP-ViT-B/16`
> * **Dataset:** `Hateful Memes`
> * **Corruption Types:**
> 	- **Image Corruptions:** `Gaussian Noise`, `Gaussian Blur`, `Random Occlusion`
> 	- **Text Corruptions:** `Random Token Masking` with [MASK] replacement
> - **Corruption Settings:** `70%` severity applied to corrupted modality during both training and testing
> - **Hyperparameters:** Same as reported in the main paper for both MoRA and DCP
>
> **Results**
>
> | Corruption Type | DCP   | **MoRA (Ours)** |
> | --------------- | ----- | --------------- |
> | Clean           | 71.47 | **75.23**       |
> | Gaussian Noise  | 71.21 | **75.07**       |
> | Gaussian Blur   | 64.73 | **67.41**       |
> | Occlusion       | 70.42 | **73.76**       |
> | Text Masking    | 64.01 | **68.49**       |
>
> **Key Insights**
> * **Corruption Resilience:** MoRA consistently outperforms DCP across all corruption types.

---

> ### Author Response · Authors · 2025-11-20
>
> > ***`Q4`: MoRA position on MM-IMDb and Hateful Memes.***
>
> A: Thank you for this question. We conduct comprehensive ablation studies on **MoRA's architectural placement** across MM-IMDb and Hateful Memes datasets to complement the UPMC-Food101 analysis in Figure 9 (Appendix).
>
> **Experimental Setup**
> * **Backbone:** `CLIP-ViT-B/16`
> * **Datasets:** `MM-IMDb`, `Hateful Memes`
> * **Missing Settings:** `70%` missing ratio, `both` modalities missing
> * **Hyperparameters:** Same as main paper
>
> **Results**
>
> | MM-IMDb | q     | k     | v     | qk    | qv    | kv    | qkv   |
> | ------- | ----- | ----- | ----- | ----- | ----- | ----- | ----- |
> | 8       | 53.07 | 53.41 | 52.27 | 51.58 | 52.65 | 52.67 | 50.85 |
> | 9       | 53.03 | 52.63 | 52.83 | 52.97 | 52.44 | 53.04 | 51.92 |
> | 10      | 52.00 | 52.08 | 52.13 | 52.59 | 52.97 | 52.72 | 52.49 |
> | 11      | 52.54 | 52.39 | 53.37 | 51.07 | 52.58 | 51.80 | 52.44 |
>
> | Hateful Memes | q     | k     | v     | qk    | qv    | kv    | qkv   |
> | ------------- | ----- | ----- | ----- | ----- | ----- | ----- | ----- |
> | 8             | 64.37 | 66.76 | 66.88 | 62.25 | 67.14 | 64.04 | 65.47 |
> | 9             | 65.72 | 67.88 | 67.61 | 66.75 | 67.96 | 66.18 | 67.22 |
> | 10            | 66.03 | 66.73 | 68.64 | 62.67 | 70.15 | 69.69 | 68.53 |
> | 11            | 66.31 | 68.32 | 68.10 | 67.68 | 68.25 | 68.70  | 68.96 |
>
> **Key Insights**
> * **Architectural Robustness:** Performance variation across all configurations is minimal, confirming MoRA's stability.
>
> ---
>
> > ***`Q5 & Q6`: Initialization and constrains of $\mathbf{S}^\mathrm{v}$ and $\mathbf{S}^\mathrm{t}$.***
>
> A: Thank you for these important technical questions. We conduct systematic experiments investigating **different initialization strategies and orthogonality constraints** for the Gram matrices.
>
> **Experimental Setup**
> * **Backbone:** `CLIP-ViT-B/16`
> * **Datasets:** `MM-IMDb`, `UPMC-Food101`, `Hateful Memes`
> * **Missing Settings:** `70%` missing ratio, `both` modalities missing
> * **Hyperparameters:** Same as main paper
> * **Initialization Strategies:**
> 	- **Gaussian (baseline):** $\mathbf{S}^{\mathrm{v}/\mathrm{t}}\sim \mathcal{N}(0,\sigma^2)$ following LoRA convention
> 	- **Zero:** $\mathbf{S}^{\mathrm{v}/\mathrm{t}}=\mathbf{0}$
> - **Orthogonal Constraint:** Add $\Vert\mathbf{S}^T\mathbf{S}-\mathbf{I}\Vert^2_2$ in the loss function
>
> **Results**
>
> |                                                   | MM-IMDb | UPMC-Food101 | Hateful Memes |
> | ------------------------------------------------- | ------- | ------------ | ------------- |
> | $\mathcal{N}(0,\sigma^2)$                         | 52.97   | 83.62        | 68.36         |
> | $\mathbf{0}$                                      | 52.83   | 83.27        | 68.28         |
> | $\Vert\mathbf{S}^T\mathbf{S}-\mathbf{I}\Vert^2_2$ | 52.73   | 83.49        | 67.95         |
>
> **Key Insights**
> * **No Special Requirements:** MoRA works effectively without complex initialization schemes, simplifying implementation and reducing hyperparameter tuning. We observed that the orthogonal constraint slightly improves training convergence speed.
>
> ---
>
> > ***`Q7`: Scaling factor.***
>
> A: Thanks for your suggestion. We have a scaling factor as in the vanilla LoRA in Eq (2) and Eq (4), and report its performance in `Figure 9 (Appendix)`, i.e., $\mathbf{W} + \eta \mathbf{S}^T\mathbf{G}\mathbf{S}$. We will add it to the main paper to improve clarity.
>
> ---
>
> **Reference**
>
> [1] Kim, Wonjae, Bokyung Son, and Ildoo Kim. "Vilt: Vision-and-language transformer without convolution or region supervision." ICML 2021.
>
> [2] Lee, Yi-Lun, et al. "Multimodal prompting with missing modalities for visual recognition." CVPR 2023.
>
> [3] Girdhar, Rohit, et al. "Imagebind: One embedding space to bind them all." CVPR 2023.
>
> [4] Lai, Kevin, Liefeng Bo, and Dieter Fox. "Unsupervised feature learning for 3d scene labeling." ICRA 2014.

---

> ### Author Response · Authors · 2025-11-27
>
> We believe that our follow-up experiment results and explanations address the concerns raised by the reviewer. Since the rebuttal window will close within one week, we respectfully ask that they consider raising their score to reflect these clarifications and improvements if the reviewer finds them satisfactory. We are happy to answer any remaining questions.

---

### Author Response · Authors · 2025-11-29

We sincerely appreciate all reviewers for their insightful and constructive feedback. We extend our special gratitude to ACs/SACs/PCs for their tremendous efforts, particularly during this period of significantly increased workload.

---

***Summary***

We present MoRA, a parameter-efficient fine-tuning method for vision-language models in missing-modality scenarios. Our innovations include introducing low-rank-based methods instead of prompt-based methods in this research area and proposing a gram-matrix-based solution to address the dimension-mismatch issue, a unique challenge in applying low-rank-based methods to this task. Our method achieves 5.24% average improvement over SOTA while using only 25.90% inference time and 0.11% trainable parameters. During rebuttal, we conducted extensive additional experiments addressing all reviewer concerns.

---

***Key Concerns and Our Responses***

We identified several main concerns raised across reviewers and addressed each with new experiments.

1. ***Generalization to Single-Stream Architectures (Reviewers `qZrT`, `fEg2`, `3Yzm`, `AxMw`***)

**Concern**: MoRA is designed for dual-encoder models (CLIP). Does it generalize to single-stream/early-fusion architectures?

**Response**: We conducted new experiments on ViLT (a backbone used in previous baselines) across three datasets.

**Key insight**: MoRA successfully adapts to single-branch architectures, matching or exceeding prompt-based baselines, while maintaining zero inference overhead.

2. ***Scalability Beyond Image and Text (Reviewers `qZrT`, `3Yzm`, `AxMw`***)

**Concern**: Can MoRA scale beyond Image and Text modalities, and even scale to 3+ modalities without exponential complexity growth?

**Response**: We demonstrated extension to 3 modalities (RGB + Depth + Text) on RGB-D Scenes Dataset v2 dataset.

**Key Insight**: MoRA outperforms baselines, confirming scalability. The Gram matrix design enables linear (not exponential) scaling with additional modalities.

---

***Additional Rebuttal Contributions***
* **Real-world Applications**: Providing more real-world applications for missing modality tasks, including Healthcare, Social Media, and Autonomous Vehicles.
* **Unaligned Encoder Experiments**: MoRA with individual ViT and Bert still outperforms baselines, confirming it enables modality interaction.
* **Partial Corruption Robustness**: MoRA maintains strong performance under image patch masking and text word dropout.
* **Hyperparameter Robustness**: Providing more analysis regarding hyperparameters shows MoRA's strong robustness across hyperparameters.
* **Initialization Robustness**: MoRA achieves strong performance across different initialization methods.

---

***Conclusion***

We believe our comprehensive rebuttal addresses all reviewer concerns with concrete experimental evidence.

---

### Meta-Review · Area_Chair_LN4m · 2026-01-02

**Summary:**

The paper initially received one negative and three positive ratings. The concerns are mostly about 1) experimental results, e.g., extension to single-branch models, other modality scenarios, hyperparameters, 2) motivation of the problem setting, 3) some technical clarifications, e.g., experimental setting.

**Reviewer Concerns:**

The authors have provided responses in the rebuttal to answer initial concerns from the reviewers. The AC took a close look at the paper, reviews, and the rebuttal. After the rebuttal, the AC finds that most questions are addressed well, especially the further clarification of technical contributions and additional experiments, e.g., extension to other modality settings and base models. While no reviewers indicated an increase in the initial rating, the AC agrees with the reviewers' overall positive feedback and hence recommends the acceptance rating, while strongly encouraging the authors to revise the paper accordingly and release the code for reproducibility.

**Reviewer Scores:**

Reviewer AxMw mentioned that the questions are addressed well without indicating the rating change, while the other three reviewers did not fully participate in the discussion.

---

### Decision · Program_Chairs · 2026-01-26

Accept (Poster)